# The Emerging Roles of Fox Family Transcription Factors in Chromosome Replication, Organization, and Genome Stability

**DOI:** 10.3390/cells9010258

**Published:** 2020-01-20

**Authors:** Yue Jin, Zhangqian Liang, Huiqiang Lou

**Affiliations:** Beijing Advanced Innovation Center for Food Nutrition and Human Health and State Key Laboratory of Agro-Biotechnology, College of Biological Sciences, China Agricultural University, No.2 Yuan-Ming-Yuan West Road, Beijing 100193, China

**Keywords:** DNA replication, chromatin interaction, transcription-independent, chromosome domain, replication-transcription conflicts, cell fate decision

## Abstract

The forkhead box (Fox) transcription factors (TFs) are widespread from yeast to humans. Their mutations and dysregulation have been linked to a broad spectrum of malignant neoplasias. They are known as critical players in DNA repair, metabolism, cell cycle control, differentiation, and aging. Recent studies, especially those from the simple model eukaryotes, revealed unexpected contributions of Fox TFs in chromosome replication and organization. More importantly, besides functioning as a canonical TF in cell signaling cascades and gene expression, Fox TFs can directly participate in DNA replication and determine the global replication timing program in a transcription-independent mechanism. Yeast Fox TFs preferentially recruit the limiting replication factors to a subset of early origins on chromosome arms. Attributed to their dimerization capability and distinct DNA binding modes, Fkh1 and Fkh2 also promote the origin clustering and assemblage of replication elements (replication factories). They can mediate long-range intrachromosomal and interchromosomal interactions and thus regulate the four-dimensional chromosome organization. The novel aspects of Fox TFs reviewed here expand their roles in maintaining genome integrity and coordinating the multiple essential chromosome events. These will inevitably be translated to our knowledge and new treatment strategies of Fox TF-associated human diseases including cancer.

## 1. An Evolutionary Overview of Fox Family Transcription Factors (TFs)

The forkhead box (Fox) family of transcription factors (TFs) spans from unicellular eukaryotes to humans, aside from plants (Figure 1). The Fox family has four members, Fkh1, Fkh2, Fhl1 and Hcm1, in *Saccharomyces cerevisiae*. It flourishes to be one of the largest classes of TFs in human. The Fox family TFs share a conserved common structurally related DNA binding domain (DBD), the forkhead domain. This domain belongs to a much larger superfamily, the winged-helix. The winged-helix/forkhead class of TFs is characterized by a 100-amino-acid monomeric DBD folded into a variant of the helix–turn–helix motif with three α helices and two characteristic large loops or so-called “wings” [1,2].

## 2. The DNA Sequence Bound by Various Fox TFs

The DNA-binding specificities of different forkhead proteins were intensively examined (Figure 2). The canonical forkhead target sequence is RYAAAYA, which is referred to as the forkhead primary (FkhP) motif. A similar variant, AHAACA, was identified during in vitro selection and protein-binding microarray experiments for several Fox proteins. It was designated as the forkhead secondary (FkhS) motif. A third motif, (G)ACGC, is called the FHL motif, which is the preferred binding site of FoxN1, N4, and Fhl1 in vitro and in vivo [4]. 

Besides various sequence specificities, the roles of Fox TFs might be regulated through additional layers such as protein homo- or hetero-oligomerization and distinct DNA binding patterns, which are discussed in Section 5 in more detail.

## 3. The Role of Fox TFs in DNA Replication

DNA replication is a fundamental process essential for all living beings. Each cell needs to accurately duplicate and distribute the whole set of genetic information into two daughter cells. DNA replication is strictly controlled in an orchestrated manner according to different stages of the cell cycle in all eukaryotes. It is crucial for genome stability maintenance, cell proliferation, and cell fate decision. During the past decades, Fox family TFs were demonstrated to play critical roles in regulating DNA replication and cell cycle through both transcription-dependent and transcription-independent mechanisms.

### 3.1. Fox TFs Regulate DNA Replication in a Transcription-Dependent Way

Early work on Fox TFs focused on their function as a general transcription factor to control gene expression to affect DNA replication.

In *Saccharomyces cerevisiae*, Fkh1 and Fkh2 proteins bind the promoters of the “CLB2 cluster”, which contains 33 genes, including CLB1, CLB2, SWI5, ACE2, CDC5, and CDC20. They act as transcription factors to ensure the cell-cycle regulated expression of these genes, then drive progression through mitosis after binding to the Cdk1 kinase [5]. 

More direct evidence has come from the role of Fhl1 in ribonucleotide reductase (RNR) gene expression. Ribonucleotide reductase catalyzes the rate-limiting step in the de novo biogenesis of deoxyribonucleotide triphosphates (dNTPs). It usually comprises a homodimer of large subunit, Rnr1, and a heterodimer of two small subunits, Rnr2 and Rnr4. Another large subunit, Rnr3, is only induced through a multi-level surveillance system when cells suffer replication stress or DNA damage [6,7]. Heterozygous deletion of FHL1 reduces transcription of RNR1 and RNR3 (but not RNR2 and RNR4). Chromatin immunoprecipitation (ChIP) shows that Fhl1p binds to the promoter regions of RNR1 and RNR3. The Δfhl1/FHL1 mutant confers a decrease in dNTP levels and an increase in hydroxyurea (HU) sensitivity. This study suggests that Fox TF, from another aspect, affects DNA replication by regulating the supply of building blocks of nucleic acids [8]. 

Similar to Fox TFs in yeast, the expression of FOXM1B is increased at the G1/S transition in regenerating liver. FOXM1B protein directly binds the CDC25B promoter region and regulates the expression of cell cycle proteins essential for hepatocyte entry into DNA replication and mitosis [9]. FOXM1 controls transcription of the mitotic regulatory genes Cdc25B, Aurora B kinase, survivin, centromere protein A (CENPA), and CENPB in both mouse embryonic fibroblasts (MEFs) and human osteosarcoma. Moreover, FoxM1 is also essential for the expression of Skp2 and Cks1. The latter two are substrate recognition subunits of the Skp1-Cullin 1-F-box (SCF) ubiquitin E3 ligase that targets p21Cip1 and p27Kip1, the CDK inhibitor (CDKI) proteins for degradation during the G1/S transition [10]. Therefore, FOXM1 deficiency leads to elevated nuclear levels of these CDKI proteins, which may account for a significant decrease in proliferating cells and an increase in apoptotic cells [11]. Cooperation of FOXM1 and AR accelerated DNA synthesis and cell proliferation by affecting CDC6 gene expression [12]. FOXO3 could form a complex with DDB1 and compete with DDB1 and PCNA interaction [13].

There is also evidence to show that FOXM1 is regulated by B-Myb, which a key TF in the cell cycle regulation of somatic cells and implicated in different types of human cancer. B-Myb is ubiquitously expressed in various cell types. However, the levels of B-Myb in embryonic stem cells (ESCs), embryonic germ cells, and embryonic carcinoma cells are over 100 times than those in normal proliferating cells. B-Myb ablated ESCs show a significantly decreased expression of FOXM1 and c-Myc. ChIP results reveal a specific enrichment of B-Myb at the FOXM1 locus in the binding site 2 region (BS2) [14]. Moreover, through a systematic screen, Anders et al. identified FoxM1 as a substrate of cyclin D1-CDK4 and cyclin D3-CDK6. FOXM1 protein is stabilized and activated after phosphorylation by CDK4/6, then plays its role in the expression of G1/S phase genes. Meanwhile, it suppresses the levels of reactive oxygen species (ROS), maintains genome stability, and protects cells from ROS-induced senescence. In conclusion, Fox TFs functions in signaling cascades and gene expression to regulate DNA replication, cell cycle progression, and cell fate decisions from yeast to human.

### 3.2. Fox TFs Regulate DNA Replication in a Transcription-Independent Way

Besides operating as TFs for transcription of the genes involved in DNA synthesis and cell cycle control, Fox family proteins were recently shown to have some direct roles in DNA replication timing. DNA replication consists of three stages, initiation, elongation, and termination. Replication initiation can be further divided into at least two steps [15,16]. First, in G1 phase, two hexameric Mcm2–7 (minichromosome maintenance 2-7, MCM) rings are loaded onto duplex origin DNA through ORC, Cdt1, and Cdc6, forming a double hexameric MCM complex called the prereplication complex (pre-RC) [17,18,19,20,21]. This step is called licensing. Second, Dbf4-dependent kinase (DDK) catalyzes MCM phosphorylation a their N-termini, which mediates the recruitment of Sld3-Sld7-Cdc45 and the assembly of the Cdc45–MCM–Sld3 (CMS) complex [15,22,23] When cells proceed into S phase, the S-phase cyclin-dependent kinases (S-CDKs) phosphorylate Sld3 and Sld2 to stimulate their interactions with Dbp11-Polε-GINS, resulting in the assembly of the preinitiation complex (pre-IC) complex [24,25]. With the help of Mcm10, it is subsequently remodeled into two active Cdc45-MCM-GINS (CMG) helicases, which eventually trigger origin unwinding and bidirectional replication [26,27,28,29,30,31,32]. 

Besides such exquisite step-by-step assembly of the replication machine in each origin, DNA replication is also conducted by a global tempo-spatial program throughout the genome. Unlike bacteria, Eukarya exploit a large number of origins ranging from ~500 in yeast to 50,000 in humans. Interestingly, origins do not fire simultaneously but follow some particular timing program [33,34,35,36].

The timing program is established in the late G1 phase, which is called timing determinant point [34]. Based on a series of studies in yeast from several groups, a limiting factor model was proposed for the determination of replication timing. The firing factors such as Sld3 and Cdc45 are available in limited amounts relative to the total number of origins in budding yeast. Overexpression of these factors often results in the advanced firing of some late origins [37,38,39]. Meanwhile, Sld3 and Cdc45 are enriched at early origins in G1 in a DDK-dependent manner [38,40]. The essential role of DDK attributes to MCM phosphorylation, which mediates the recruitment of Sld3-Cdc45 through direct association with a basic patch motif within Sld3 [15,23].

Fkh1 is first implicated in modulating the late-origin firing and heterochromatin structure of the mating-type locus in a genetic study in budding yeast [41]. Its exact role in DNA replication was not known until seminal studies from Aparicio and his colleagues [42]. By using a quantitative genome-wide BromodeoxyUridine immunoprecipitation-sequence (BrdU-IP-seq), they uncovered that Fkh1 and Fkh2 somewhat redundantly determine a subset (~30%) of early origin firing. The bioinformatic analysis predicted that these origins contain more than one Fkh binding site (FBS), which was validated by the ChIP results. FKH1 or FKH2 overexpression advances the initiation timing of many origins throughout the genome, resulting in a higher total level of origin firing in the early S phase. On the other hand, deletion of *FKH1* and *FKH2* or their binding sites proximal to Fkh-activated origins results in delayed activation of these origins. As a consequence, other origins, referred to as Fkh-repressed origins, become activated in the absence of *FKH1* and *FKH2*, likely due to reduced competition from Fkh-activated origins for dose-limiting replication initiation factors.

In an independent study, through high-throughput yeast two-hybrid screens, Fang et al. identified two novel Dbf4 interactors: Fkh1 and Fkh2 [23]. ChIP analysis showed that Fkh TFs are required for the enrichment of Dbf4 in a subset of early origins in G1 but not for the recruitment of pre-RC components such as ORC and MCM. Next, by using the purified proteins and the biotin-labeled origin DNA, they reconstituted the recruitment of Dbf4 to these early origins in vitro. The minimal requirements of Dbf4 recruitment are Fkh and an origin bearing FBS (FBS^+^). This indicates that the pre-RC assembly is not a prerequisite for Dbf4 recruitment. These findings demonstrated that Dbf4 is barely able to bind origins per se, and Fkh TFs are sufficient to recruit Dbf4 to the FBS^+^ group of origins.

Very interestingly, Tomoyuki and colleagues found that Dbf4 is recruited to the early origins near centromeres through another mechanism [16]. Through GFP-labeled PCNA, a sliding clamp for DNA polymerases, they noticed that replication foci initially localize in the spindle pole body (SPB, equivalent to the centrosome in metazoa). As with PCNA, initiation factors Cdc7-Dbf4 and Sld3-Sld7 localize at centromeric regions in telophase to G1 phase. In the absence of Ctf19, a component of the kinetochore complex (also called COMA, Ctf19p-Okp1p-Mcm21p-Ame1p), the formation of Sld7 foci near SPB is diminished. ChIP-qPCR analysis of *ctf*1*9*∆ cells revealed a normal enrichment of Dbf4 at early origins such as ARS606 or ARS607 but a reduced binding of Dfb4 to early origins located in peri-centromeric regions. Collectively, these two independent studies elucidated that Dbf4 is preferentially recruited to early origins through distinct mechanisms in a chromatin context-dependent manner. 

The very C-terminal 50 amino acid of Dbf4 mediates its interaction with Fkh1 and Fkh2. The interaction-defective mutant, dbf4ΔC, phenocopies fkh1Δ alleles in terms of origin firing. More importantly, the direct fusion of the DNA-binding domain (DBD, also called forkhead) of Fkh1 to Dbf4ΔC fully restores the Fkh-activated origin firing. In control, a fusion of DNA-binding defective forkhead mutant with Dbf4ΔC results in no rescue at all. These findings convincingly demonstrated that DNA-binding activity but not the transcription activation activity of Fkh TFs is necessary for the recruitment of Dbf4. In other words, Fkh TFs determine DNA replication timing through direct physical interaction with DDK and are utterly independent on their transcription roles. 

Intriguingly, genome-wide replication profiles show that Dbf4 C-terminal fusion with either forkhead or an epitope interferes with the early replication of pericentromeric origins. It was noticed that the addition of a C-terminal tag may specifically abolish the interaction of Dbf4 with Ctf19. In addition to its role as an essential regulatory subunit of DDK, Dbf4 interacts directly with Sld3, which may be attributed to the direct recruitment of these downstream limiting factors. This represents the first clue that Dbf4 may play a direct role in regulating DNA replication in a DDK-independent way. 

These studies depict how early origins in the different chromosome context compete for DDK, the upstream rate-limiting factor in determining the replication timing program in G1.

### 3.3. Rif1-PP1 Phosphatase Negatively Regulates Replication Initiation to Compete for Fox TFs

Contrary to the stimulatory role of Fkh1 and Fkh2 in origin firing, Rif1-mediated PP1 phosphatase inhibits replication initiation through reversing MCM phosphorylation. Rif1 (RAP1-interacting factor) was originally identified as a telomere-binding factor to regulate the telomere length in yeast [43]. The Masai group first discovered a critical role of Rif1 replication timing in human cells [44]. Rif1 colocalizes specifically with the mid-S replication foci and establishes the mid-S replication domains that are restrained from being activated at early-S-phase. Rif1 prolongs the embryonic S phase at the Drosophila mid-blastula transition [45]. The Donaldson group showed that deletion of *RIF1* increases the proportion of hyperphosphorylated Mcm4 and partially compensates for the limited DDK activity in a temperature-sensitive (ts) mutant of the catalytic subunit of DDK, *cdc7-1,* in yeast [46]. Rif1 has two conserved N-terminal motifs, RVxF and SILK, which directly associate with Glc7, the sole protein phosphatase 1 (PP1) in budding yeast [47]. Mutation of these domains increases MCM phosphorylation and thus suppresses the growth defects of *cdc7-4* and *dbf4-1* mutants. ChIP results confirmed that the Rif1–PP1 interaction is necessary for PP1 enrichment in late origins in both *S. cerevisiae* and *S. pombe*. After docking onto the pre-RC through Rif1, PP1 then reverses the MCM phosphorylation carried out by DDK and represses origin firing. Rif1 can also mediate MCM dephosphorylation at replication forks, and the stability of dephosphorylated replisomes strongly depends on Chk1 activity in animals [48].

Interestingly, MCM phosphorylation is regulated by additional mechanisms [49]. After loading, MCM is SUMOylated, which peaks in G1 and declines during S-phase, then rises again in the M phase. DDK is required for the S-phase decline of MCM SUMOylation. SUMOylation of Mcm6 increases its interaction with Glc7, promoting the MCM dephosphorylation. 

Besides MCM, RIF1-PP1 protects the origin-binding ORC1 from premature phosphorylation and consequent degradation by the proteasome in G1 [50]. Meanwhile, Rif1 is to counteract DDK phosphorylation of Sld3 as well. Very recently, Javier Garzón et al. showed that human RIF1-PP1 protects nascent DNA from over-degradation by DNA2 at stalled replication forks and limits phosphorylation of WRN at sites implicated in resection control [51].

Bringing things full circle, PP1/Rif1 interaction is downregulated by the phosphorylation of Rif1, most likely by CDK/DDK [52]. Indeed, putative conserved DDK and CDK phosphorylation sites were found adjacent to the protein phosphatase 1 docking domains in both *S. cerevisiae* and *S. pombe* Rif1. When nine putative DDK or CDK sites in Rif1 are changed to alanine, the temperature sensitivity of cdc7-1 is enhanced, while changing them to mimic phosphorylation (aspartic acid) has the opposite effect.

In conclusion, MCM loading and pre-RC assembly occur in all origins throughout the genome, which means that all of them are licensed for replication. However, the replication timing and the efficiency of each origin are determined by the phosphorylation of MCM. This critical event is precisely controlled by protein kinase DDK and phosphatase PP1, which are mediated by Fox TFs (or COMA at pericentromeres) and Rif1, respectively (Figure 3a).

## 4. The Role of Fox TFs in Origin Clustering, Relocalization and Replication Factories

Besides helping “the early birds catch the worm”, which means that Fox TFs recruit the limiting initiation factors to early origins [39], there is a great deal of evidence that 3D chromatin structure represents another dimensional regulation of replication timing. For instance, even when the replication limiting factors are overexpressed, RPD3 needs to be knocked out to activate the dormant origins [37]. There are excellent reviews on epigenetic determinants and dynamic chromosome organization of replication timing [18,34,35,53,54,55,56].

Fox TFs were first found to be required for the clustering of early origins in G1 by the Aparicio group [42]. They observed a non-random distribution of Fkh-activated and -repressed origins. Origins of each class—not just limited to CEN- and TEL-proximal ones—often cluster linearly along the chromosome. Indeed, 4C (chromosome conformation capture-on-chip) reveals both intrachromosomal and interchromosomal interactions of Fkh-activated origins in G1 phase in an Fkh-dependent manner. Fox TFs do not participate in the formation of topologically associating domains (TAD), but they mediate the long-range interactions of origins at TAD boundaries [57]. On the other hand, the involvement of telomere-binding proteins such as Rif1 as global regulators of the replication timing of subtelomeric and many internal origins implies a role for these proteins in organization or localization of origins within the nucleus. Palmitoylation of Rif1 regulates the association of telomeres with the nuclear periphery, suggesting that palmitoylated Rif1 anchors chromatin to the nuclear membrane [58]. Origins locate at the nuclear periphery often replicate late, whereas early origins are often observed in the nuclear interior during G1 [59,60]. When FBS late origin ARS501 in the subtelomeric region of chromosome V-R is replaced by Fkh-activated origin ARS305, the “new” origin (*ARS305V-R*) loses early-firing in *fkh1*∆*fkh2*∆ cells [61]. If Fkh1 is induced in G1-phase, *ARS305V-R* regains early replication in the succeeding S phase. Using this Fkh1-induced origin activation system, Zhang et al. recently labeled origin *ARS305V-R* with tetO/TetR-Tomato [62]. They observed this subtelomeric origin re-positions from the nuclear periphery to the interior upon Fkh1-induction in G1 and replicates early in S. This phenomenon, called Fkh1-dependent origin relocalization, disappears in *cdc7-4* and*cdc45-1* ts mutants as well as in Fkh interaction defective mutant dbf4∆C. Moreover, an MCM4-14D mutant mimicking the phosphorylation status by DDK can bypass the requirement of DDK. Therefore, the origin mobility depends on MCM phosphorylation by Fkh1-mediated DDK and subsequent Cdc45 loading [62]. These seminal studies argued that the replication of eukaryotic chromosomes is organized temporally and spatially in a four-dimensional manner within the nucleus through Fox TFs and epigenetic elements. Origin clustering enables cooperativity between origins in the recruitment of limiting initiation factors Dbf4, Sld3, and Cdc45. Such assemblages inevitably fit for the concentration of replication factors and DNA synthesis, i.e., the observed replication foci, thus representing an in vivo evidence to support the theory of replication factories (Figure 3b).

## 5. Dimerization of Fkh Contributes to Origin Clustering and Dynamic Localization

Precise spatial and directional arrangement of Fkh1/2 sites is crucial for the efficient binding of the Fkh1 protein and for early firing of the origins [63]. Both Fkh1 and Fkh2 harbor a domain-swapping motif (DSM) that allows for either homo- or hetero-dimerization [64]. which underlies and accounts for the observed origin clustering. Crystal structures of the human FoxP2 and FoxP3 forkhead DNA-binding domains also provide the possibility to form domain-swapped dimers, which may “catch” two origins [65,66]. *fkh1* with DSM mutated (*fkh1*-dsm) cannot dimerize, which binds origins in vivo but fails to cluster them. Therefore, Fkh1/2 dimers perform a structural role in the spatial organization of chromosomal elements with functional importance. However, such mutations pose a subtle effect on their transcription function, suggesting that the different binding patterns of Fkh determine their distinct roles in transcription and replication/chromatin organization. Interestingly, Fkh1/2 bind strongly at their transcriptional target genes of the CLB2 group. In comparison, their binding to replication origins and recombination enhancer is relatively weak and dynamic [64].

## 6. Summary and Prospects

Fox TFs are well-known to maintain genome stability through participation in the process of DNA damage response and repair. Here, we summarize their crucial physiological roles in DNA replication, providing a new aspect of this highly conserved family of TFs because they participate in most, if not all, essential biological processes critical for cell fate decisions, including cell cycle progression, cell proliferation, cell renewal, cell differentiation, cell migration, and cell survival. Additionally, the expression of Fox TFs is frequently upregulated in many types of tumors. For example, the upregulation of FOXM1 expression is an early event during cancer initiation, progression, invasion, metastasis, and drug resistance [67,68]. Therefore, the novel functions of Fox TFs in DNA replication and chromosome organization will inevitably shed new light on the related genome-instability diseases. 

Despite the rapid-growing knowledge of Fox TFs in the regulation of chromosome replication and structure, there are still some key scientific questions ahead, for example:Do FOX TFs determine replication timing in higher eukaryotes? Which FOX TFs are required?In addition to a critical DDK regulator, are there other transcription-independent roles of FOX TFs in DNA replication?Are there more FOX TFs participating in regulating DNA replication and maintaining genomic stability?Do FOX TFs participate in other chromosome processes such chromosome segregation?Are FOX TFs involved in high-order chromosome organization in higher eukaryotes?

## Figures and Tables

**Figure 1 cells-09-00258-f001:**
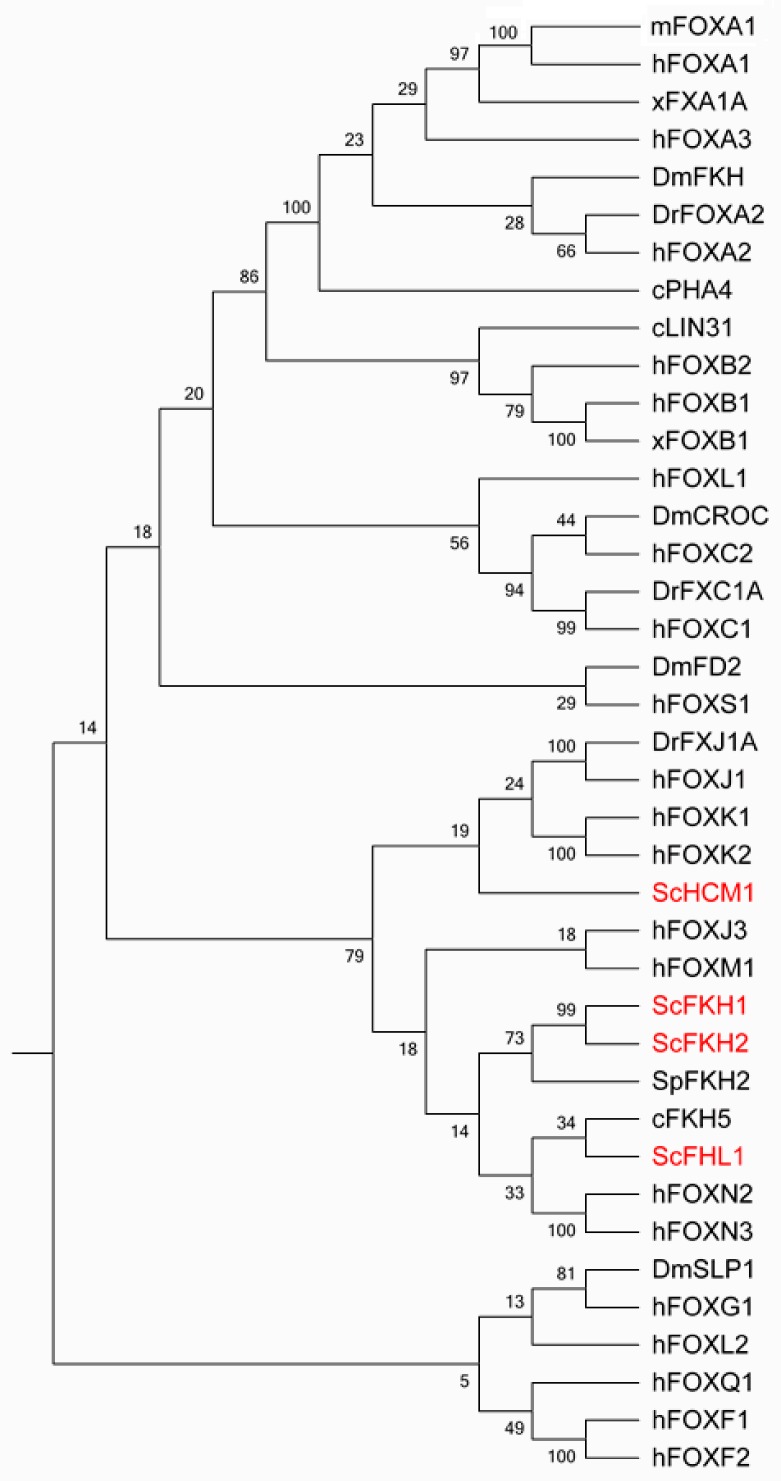
Phylogeny tree of representative forkhead box (Fox) transcription factors (TFs) from the primary model eukaryotic organisms based on the amino acid sequence of the forkhead domain. The evolutionary history was inferred using the neighbor joining method in MEGA7. The bootstrap consensus tree inferred from 100 replicates was taken to represent the evolutionary history of the taxa analyzed. Branches corresponding to partitions reproduced in less than 50% bootstrap replicates are collapsed. The percentage of replicate trees in which the associated taxa clustered together in the bootstrap test (100 replicates) are shown next to the branches. The evolutionary distances were computed using the Poisson correction method and are in the units of the number of amino acid substitutions per site [3]. Sc: *Saccharomyces*
*cerevisiae*, Sp: *Schizo*s*accharomyces pombe*, c: *Caenorhabditis elegans*, Dr: *Danio rerio*, Dm: *Drosophila melanogaster*,m: *Mus musculus*, x: *Xenopus Laevis*, h: *Homo sapien*.

**Figure 2 cells-09-00258-f002:**
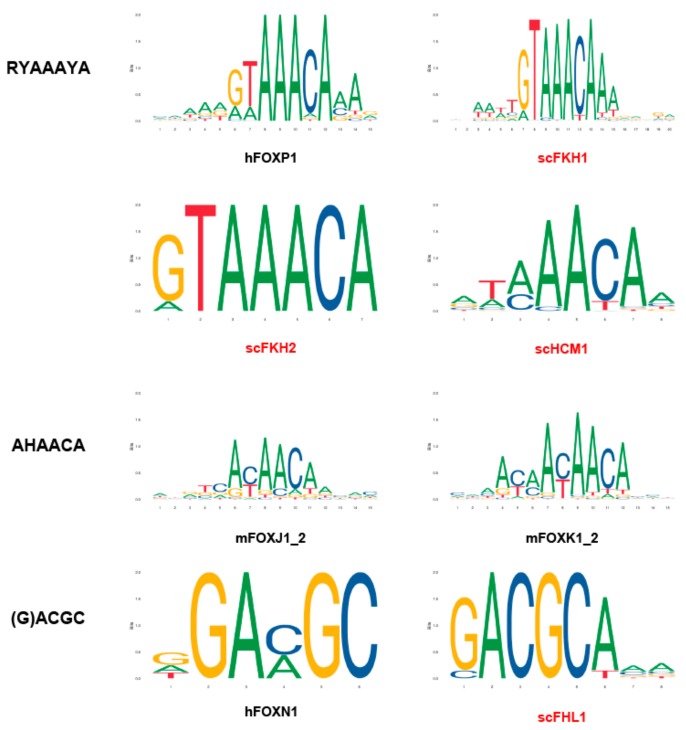
The DNA motifs preferentially bound by Fox TFs. TF binding profiles are obtained through JASPAR, an open-access database (http://jaspar.genereg.net) {Fornes, 2020 #174}.

**Figure 3 cells-09-00258-f003:**
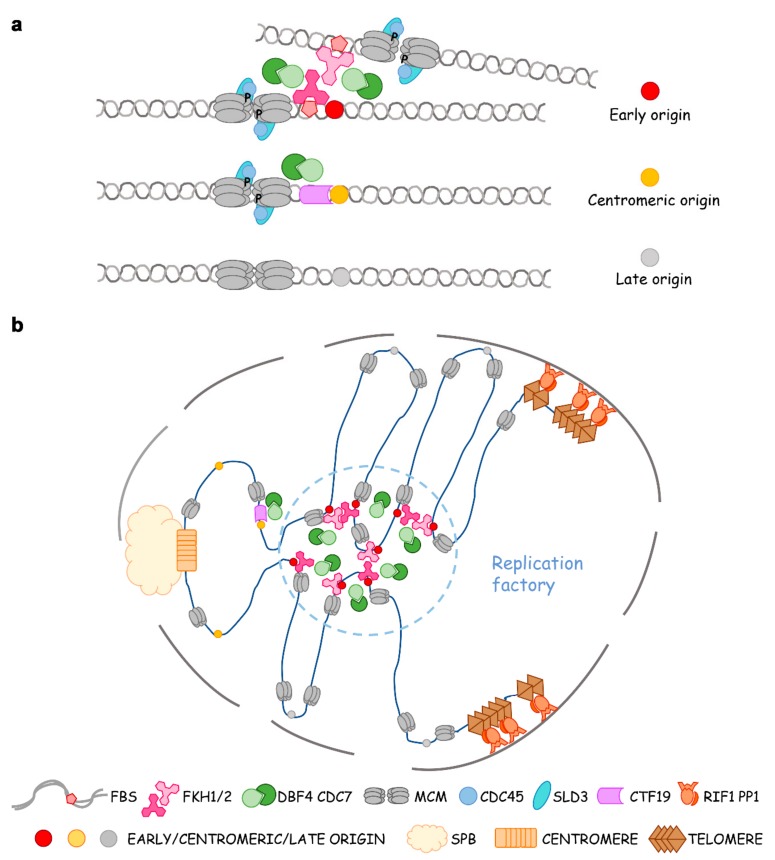
Fox TFs determine the global replication timing program through preferential recruitment of limiting factors, origin clustering, repositioning, and making replication factories. (**a**) Dbf4-dependent kinase (DDK is recruited to chromosomal arms and pericentromeric origins via Fkh1/2 and Ctf19p-Okp1p-Mcm21p-Ame1p (COMA), respectively. Dbf4 recruitment defines an upstream rate-limiting step in determining replication timing in yeast. A subset of early origins is bound by Fkh1 or Fkh2, which recruits Dbf4 and downstream limiting factors Sld3-Cdc45 in late G1. In pericentromeric origins, Ctf19 recruits DDK to compete for limiting factors. Late or dormant origins need to wait for the cycled use of DDK and other firing factors. (**b**) Model of Fkh-dependent origin clustering, assemblage of replication components, and formation of replication factories.

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
