# Peer review of "The Emerging Roles of Fox Family Transcription Factors in Chromosome Replication, Organization, and Genome Stability"

_cells, 2020, doi:10.3390/cells9010258_

Round 1
Reviewer 1 Report
In this review, Jin et al summarized our recent understanding of the emerging role of the forkhead box (Fox) family transcription factors in DNA replication regulation, including the initiation and spatial organization of replication in the context of 3D chromatin structure.
Specific Concerns:
The title contains “genome-instability diseases”, but there is very little discussion of Fox TFs in the context of human diseases in the main text. The authors may consider rephrasing the title. The authors may explain the meanings of the numbers in the Figure 1. The authors may re-organization Figure 2 to make it consistent with the description in the main text. In the section 3.3, the authors discussed how Rif1-PP1 phosphatase negatively regulates replication initiation, but barely touched how Fox TFs are involved in this regulation. Figure 3A is very confusing and it is hard to find the various origins based on the legends. The authors may consider rephrasing the subtitle of Section 4.
Author Response
point to point response letter
Specific Concerns:
The title contains “genome-instability diseases”, but there is very little discussion of Fox TFs in the context of human diseases in the main text. The authors may consider rephrasing the title.
A: Good suggestion, changed as “genome stability”.
The authors may explain the meanings of the numbers in the Figure 1.
A: Added in the new legend.
The authors may re-organization Figure 2 to make it consistent with the description in the main text.
A: See new Figure 2 and also updated reference.
In the section 3.3, the authors discussed how Rif1-PP1 phosphatase negatively regulates replication initiation, but barely touched how Fox TFs are involved in this regulation.
A: We rephrased the subtitle and the beginning to make it more cohesive.
Figure 3A is very confusing and it is hard to find the various origins based on the legends.
A: Thanks for pointing out, origin was missed in the previous Fig 3A.
The authors may consider rephrasing the subtitle of Section 4.
A: rephrased as “The role of Fox TFs in origin clustering, relocalization and replication factories”.
Reviewer 2 Report
Comments on ‘The emerging role of Fox family transcription factors in chromosome replication, organization and genome-instability diseases’ by Yue Lin et al. This is a very interesting subject and this review will be of broader interest.
The authors review the role of Fox proteins in DNA replication timing (Chapter 3) and chromosomal organization (Chapter 4). These chapters should be carefully rewritten in order to provide a better understanding to the reader.
General comments: Important! Throughout the text present tense must be replaced by past tense. For example see line 47: A similar variant, AHAAC, is identified……
Correct would be: A similar variant, AHAAC, has been identified……
The authors discuss the role of Fox proteins in different organisms. They should clarify along the text which results were derived from each organism.
Specific comments:
Line 58: Capter 3; The title ‘Fox TFs in DNA replication’ should be changed to something like: ‘The role of ……’ or ‘Fox TFs regulate….’. The titles of chapter 4 also needs to be revised.
Line 67: ‘In the early days’ should be should be replaced by ‘Early work on…..’
Chapter 3.1; line 105: It should be explained how the FOXM1 protein interferes with ROS production.
Chapter 3.2; lane 111: ….direct roles in DNA replication. The sentence should be changed to ….. e.g. direct roles in DNA replication timing.
Line 150: …: Fhk1 and Fhk2. Reference is missing
Line 159: Through fluorescence-labeled PCNA…. The authors should clarify this experiment.
Line 163:, …..Sld7 foci near SPB is diminished. Change to: …..the formation of Sld7 foci near SPB is diminished.
Line 164: ChIP qPCR shows that …….Dfb4….. This sentence needs to changed to: ChIP qPCR analysis of ctf9∆ cells revealed a normal enrichment of Dbf4 at early origins such as ARS606 or ARS607, but a reduced binding of Dfb4 to early origins located in peri-centromeric regions.
Line 175:….recruitment of Dbf4. The author should clarify this sentence.
9: Line 186: The title of Chapter 3.3 should be related to Fox proteins.
10: Line 94: It needs to be defined what is Cdc7.
11: Lines 203 to 223. This part of the text summarizes work of the A. Donaldson lab and needs to be rewritten.
Line 225: Besides helping “the early birds catch the worm”….Please clarify this sentence.
Line 249: …. disappears in cdc7-4, cdc45-1 ts….. Please clarify if this refers to single of double mutants.
Line 254: The authors should define better what they mean with ‘four-dimensional manner’ and ‘epigenetic elements’. See also 3D and 4D organization mentioned along the text.
Line 285: Fox TFs…..though DNA damage response and repair. The authors should clarify how the Fox TFs are linked to DNA damage and repair, or if they mean ‘repair of DNA damage’.
Line 300: ‘More Fox TFs….’ The authors should clarify what the mean.
Line 301: The authors should clarify if the mean cellular segregation, chromosome segregation or both.
Figure 3 a. I could not identify Early, Late or Centromeric origins in the printout of the figure.
Figure 3 b. The authors should draw a bubble structure between the MCM proteins in order to indicate replicating DNA.
Author Response
The authors review the role of Fox proteins in DNA replication timing (Chapter 3) and chromosomal organization (Chapter 4). These chapters should be carefully rewritten in order to provide a better understanding to the reader.
A: Thanks for pointing out, these two chapters were rephrased.
General comments: Important! Throughout the text present tense must be replaced by past tense. For example see line 47: A similar variant, AHAAC, is identified……
Correct would be: A similar variant, AHAAC, has been identified……
A: Thanks for pointing out, corrected as suggestion throughout the whole text.
The authors discuss the role of Fox proteins in different organisms. They should clarify along the text which results were derived from each organism.
A: Organisms were indicated for clarity.
Specific comments:
A: We really appreciate all these specific comments. All of them are corrected as suggestion, except the very last one.
Figure 3 b. The authors should draw a bubble structure between the MCM proteins in order to indicate replicating DNA.
A: This figure depicts the events at the timing determining point, which is the moment right before the origin firing.
Line 58: Capter 3; The title ‘Fox TFs in DNA replication’ should be changed to something like: ‘The role of ……’ or ‘Fox TFs regulate….’. The titles of chapter 4 also needs to be revised.
Line 67: ‘In the early days’ should be should be replaced by ‘Early work on…..’
Chapter 3.1; line 105: It should be explained how the FOXM1 protein interferes with ROS production.
Chapter 3.2; lane 111: ….direct roles in DNA replication. The sentence should be changed to ….. e.g. direct roles in DNA replication timing.
Line 150: …: Fhk1 and Fhk2. Reference is missing
Line 159: Through fluorescence-labeled PCNA…. The authors should clarify this experiment.
Line 163:, …..Sld7 foci near SPB is diminished. Change to: …..the formation of Sld7 foci near SPB is diminished.
Line 164: ChIP qPCR shows that …….Dfb4….. This sentence needs to changed to: ChIP qPCR analysis of ctf9∆ cells revealed a normal enrichment of Dbf4 at early origins such as ARS606 or ARS607, but a reduced binding of Dfb4 to early origins located in peri-centromeric regions.
Line 175:….recruitment of Dbf4. The author should clarify this sentence.
9: Line 186: The title of Chapter 3.3 should be related to Fox proteins.
10: Line 94: It needs to be defined what is Cdc7.
11: Lines 203 to 223. This part of the text summarizes work of the A. Donaldson lab and needs to be rewritten.
Line 225: Besides helping “the early birds catch the worm”….Please clarify this sentence.
Line 249: …. disappears in cdc7-4, cdc45-1 ts….. Please clarify if this refers to single of double mutants.
Line 254: The authors should define better what they mean with ‘four-dimensional manner’ and ‘epigenetic elements’. See also 3D and 4D organization mentioned along the text.
Line 285: Fox TFs…..though DNA damage response and repair. The authors should clarify how the Fox TFs are linked to DNA damage and repair, or if they mean ‘repair of DNA damage’.
Line 300: ‘More Fox TFs….’ The authors should clarify what the mean.
Line 301: The authors should clarify if the mean cellular segregation, chromosome segregation or both.
Figure 3 a. I could not identify Early, Late or Centromeric origins in the printout of the figure.
Figure 3 b. The authors should draw a bubble structure between the MCM proteins in order to indicate replicating DNA.